# Fire and Drought Affect Multiple Aspects of Diversity in a Migratory Bird Stopover Community

**DOI:** 10.3390/biology14060597

**Published:** 2025-05-24

**Authors:** Jenna E. Stanek, Chauncey R. Gadek, Sarah E. Milligan, Brent E. Thompson, Keegan A. Tranquillo, Laura L. Trader, Charles D. Hathcock, Stephen M. Fettig

**Affiliations:** 1Los Alamos National Laboratory, Los Alamos, NM 87545, USA; cgadek@lanl.gov (C.R.G.);; 2Department of Biology, University of New Mexico, Albuquerque, NM 87131, USA; 3Bandelier National Monument, Los Alamos, NM 87544, USA; sarah_milligan@nps.gov (S.E.M.); laura_trader@nps.gov (L.L.T.); 4U.S. Fish and Wildlife Service, Sacramento, CA 95825, USA

**Keywords:** bird banding, drought, community ecology, southwestern United States, wildfire response

## Abstract

Drought and high-severity wildfires significantly affect bird communities, including migratory stop-over populations. These disturbances are interconnected, influencing ecosystems over different timescales. Using fall bird-banding data, this study examined how wildfire and drought shape avian relative abundance, taxonomic, functional, and phylogenetic diversity. After the fire, an increase in migratory insectivores contributed to a less phylogenetically diverse community. Some community changes appeared temporary, suggesting a return to pre-fire conditions over time. However, varying monsoonal conditions showed a lasting association with functional and phylogenetic diversity, emphasizing the importance of disentangling the combined effects of short-term disturbances (fire) and long-term environmental pressures (drought). While bird populations can recover from fire-driven habitat changes, persistent drought may continue to impact community composition.

## 1. Introduction

Drought and wildfire can drive substantial shifts in species distributions and patterns of habitat use worldwide [1]. Species’ life history traits, such as migratory strategy, can buffer species from disturbances or increase their vulnerability [2]. In the southwest of the Unites States of America, intense and persistent drought—coupled with historical fire suppression—has increased wildfire frequency, extent, and severity [3,4,5]. Both drought and high-severity, stand-replacing wildfires can have notable impacts on ecosystems and their avian communities [6,7,8,9,10].

Depending on its intensity and/or frequency, wildfire can have immediate and substantial impacts on the composition and demographics of avian communities [9,11,12]. Steel et al. [12] found prominent reductions in avian community richness within severely burned habitat; however, many studies have found nonlinear abundance trends in time series data that include pre- and post-fire observations that describe an initial decrease in density and abundance followed by rapid increases that can attenuate over time [9,11,13,14]. Unlike the relatively short-term effects of fire, drought can have long-term and disproportionate impacts among species and in different habitats [7,10]. Although the response to drought can vary among avian species that have differing life histories and behavioral traits, the impacts of drought are capable of imposing sizable shifts in abundance, richness, and composition on avian communities [7]. For example, decreases in avian insectivore abundance have been repeatedly recorded during periods of drought [6,15,16,17].

An inextricable link exists between drought and wildfire, each operating across different timescales. Researchers are increasingly recognizing impacts on the dynamic interaction between drought and wildfire, describing variable community responses to fire during times of drought [9,17]. The ecosystem-resetting effects of fire can alter a landscape’s response to drought. Depending on their taxonomic makeup, post-fire successional vegetative communities can create novel microclimates that buffer against the effects of drought [18]. The constituent members of avian communities are affected differently by drought and fire, revealing patterns that suggest restructuring along multiple axes of biological organization [15,19].

Migration is a critical time in a migratory bird’s life cycle, when they are pushed to their physiological limits [20] and experience the highest annual mortality rates [21,22], especially during droughts [8,16]. Species’ migratory strategies can dramatically impact their vulnerability to disturbance-induced mortality [2]. Avian communities typically comprise resident, breeding, and transient species, whose relative proportions vary throughout the year. In this context, wildfire can be expected to more severely alter the composition of migrating transient species if they have the ability to select unburned habitat. Conversely, resident communities can be more constrained by local adaptation, low dispersal tendencies, and site fidelity and, therefore, resist disturbance-induced shifts. Most research on the compound impacts of drought and wildfire on avian communities has focused on the breeding season—when bird communities are at their most stable and transient individuals are rare [7,8,9,11,14]. Far less research is available on how fire and drought in the monsoon season impact communities during migration. Compared with breeding communities, migratory communities are composed of many transient individuals that shift with phenological sequences of species arrival and departure [23,24]. Migration has been theorized to be an evolved response to escape low food availability or harsh climatic conditions during the nonbreeding season [25,26,27,28]. Changes in resource availability or other habitat conditions along migratory routes can, therefore, influence the selection of migration stopover sites. One recent fall migration study found increases in avian abundance and decreases in body fat associated with the number of acres burned [29], suggesting that a bird’s migration pathway and health can be severely affected by wildfires. Evaluating avian community responses between the migratory and resident components of a bird community across periods of drought interrupted by wildfire can elucidate unobserved ecological processes and improve our understanding of the relative risk wildfire and drought pose to birds that have variable migratory strategies.

Environmental disturbances can drive shifts in functional diversity through range expansion or contraction, phenological changes, local extinctions, or colonizations [30,31]. Local extinction and emigration from recently burned habitat can lower functional diversity [32,33]; however, as documented in plant communities, some measures of functional diversity can rapidly increase following the initial impacts of wildfire [34]. The dynamism in community function following wildfire is understudied and can enhance descriptions of community change and the ecosystem impacts from wildfire.

In the summer of 2011, a record-setting wildfire punctuated a fall bird-banding effort of more than 5 years at Bandelier National Monument in Sandoval County, New Mexico. The wildfire was the largest in New Mexico history at the time and was exacerbated by a persistent and ongoing drought in the region. Here, we use this fall bird-banding dataset to compare the relative impact of wildfire and drought during the monsoon season on community composition between migratory and resident species in an avian community from taxonomic, functional, and phylogenetic perspectives. This study highlights species-specific responses that underlie functional measures of community diversity and ties them to processes of ecological reorganization. Examining avian community change through multiple ecologically relevant axes can offer insight into the relative ability of migratory and resident communities to recover, persist, or reassemble in the face of persisting drought and ever-increasing, large-scale wildfires in the western United States.

## 2. Materials and Methods

### 2.1. Study Area

The fall migration monitoring site, Upper Alamo in Bandelier National Monument, is located at the Alamo Boundary trailhead (35.834 N, 106.608 W) in the northwestern area of Bandelier in the state of New Mexico in the United States of America. The site is a mixed-conifer forest with dispersed meadows. In addition, the bird-banding station is located at high elevation (2718 m) in a national monument that starts below 1676 m near the Rio Grande and rises in elevation westward from there. The habitat changed substantially when the Las Conchas wildfire started on 26 June 2011, and burned a total area of 64,246 ha, much of which occurred in the park. The fire was a severe, stand-replacing fire that burned 0.40 ha per second for the first 8 h. The pre-fire forest was a mature, mixed-conifer forest composed primarily of ponderosa pine, Douglas fir, white fir, limber pine, and aspen of varying size classes. Before the fire, Rocky Mountain maple, spruce, and piñon were also present. The successionally diverse mixed-conifer forest was replaced with early successional species, primarily quaking aspens and young fir trees. Remnant, dead snags and a few remaining mature ponderosa pine and white fir trees survived the fire.

### 2.2. Bird Captures

The fall migratory bird-banding station was started with the goal of learning about the species diversity and quantity of birds that use this high-elevation location during their migration period—a critical part of a bird’s annual life cycle. These data generate useful information about the status and trends [35] of birds that migrate through Bandelier National Monument. All work performed with live wild birds followed published guidelines for the use of wild birds in research [36]. The bird-banding station consisted of 20 standard, 30 mm mesh nylon mist nets that measure 12 m long by 2.5 m high. During the study, net locations were not changed. All nets were operated from August through October. Banding operations at the station were performed twice per week from the beginning of August to mid-October in 2005, 2006, and 2008 through 2021. The number of banding sessions per year varied due to weather or lack of personnel and ranged from 16 to 32 sessions with an average of 22 sessions per year. Each banding day, nets were open at sunrise to 12 pm at the latest. Net hours were recorded within 10 min of open and close times and averaged 2081 net hours per year. The birds per 100 net hour value was used to standardize the uneven effort over the course of the study.

We identified, aged, sexed—based on Pyle [37], weighed, measured, fat-scored, and checked all birds for signs of molt.

### 2.3. Vegetation Data

We used data from a separate vegetation monitoring project and, therefore, had limited vegetation data available for the nearby area for pre- and post-fire years. We selected data from 16 plots located within 1.6 km of the fall bird-banding site. The pre-fire data (from 2008 and 2009) and the post-fire data (from 2019 and 2020) included live tree density by diameter at breast height (DBH) category and shrub density. Data were also collected on tree and shrub species. The DBH categories were sapling = 2.5–15.0 cm, small trees = 15.1–30.0 cm, medium trees = 30.1–45.0, and large trees > 45.1 cm.

Additionally, we used normalized difference vegetation index (NDVI) which is a measure of the density of green vegetation, to assess changes in vegetation to characterize habitat change. We downloaded September NDVI values for the centroid coordinates of the banding site during the study period at 250 m resolution (Landsat Spectral Indices products courtesy of the United States Geological Survey Earth Resources Observation and Science Center). We chose September NDVI values because we felt it best reflected the greenness during much of the banding session each year.

### 2.4. Statistical Analyses

To standardize variable levels of effort for each year, we calculated capture rates for each species as birds per 100 net hours [38]. A net hour is a unit of measure used to calculate the amount of time that nets are open. For example, one net that is open for 1 h is equal to one net hour. For the full 16-year analysis, we excluded same-year recaptures because we wanted to compare avian species among the years to assess community changes pre- and post-fire. Additionally, we grouped subspecies into species categories to standardize the data for further analyses. For example, gray-headed juncos and Oregon juncos were considered dark-eyed juncos for this dataset. We assigned species to three categories based on Birds of the World species accounts [39], local breeding occurrence data, and expert knowledge: (1) resident birds that use the habitat in the surrounding area year-round, (2) migratory birds that represent transient individuals that use the habitat as a stopover site, and (3) breeders that represent long-distance migrants suspected or known to breed at and around the study site (Appendix A). To evaluate how resident, breeding, or migratory species may differ in their response, we performed subset analysis with the three assigned categories.

We evaluated taxonomic change in the migratory and resident components of the avian community from 2005 to 2021 (omitting 2007 due to a mismatch in effort) over our study period, using species richness and relative abundance based on birds per net hour. The year of the fire (2011) was not considered in pre- and post-fire comparison tests. Species richness is calculated as the number of species in a sample (year, in our case). We included Palmer Drought Severity Index (PDSI) values as a measure of the annual monsoon season (July–September) [40]. Although the average annual value of PDSI was positively correlated to the annual monsoon season values (r = 0.585, *p* = 0.017, n = 16), we decided that the annual monsoon values more directly dictate resources pulses that would affect habitat usage during fall migration (mid-August–mid-October) and eliminates noise from combining PDSI variation during the previous spring and winter months. We compared vegetation variables, species richness, PDSI, and NDVI pre- and post-fire medians and tested for differences between the groups using Wilcoxon signed-rank tests due to nonparametric data distributions. We used nonparametric median-based, linear models (α = 0.05) as a robust way to evaluate relationships between diversity metrics and climatic or habitat variables with our small dataset where we could not confidently assume normally distributed response variables.

We also used nonmetric multidimensional scaling (NMDS) with a Jaccard dissimilarity matrix to assess changes in the presence and absence of different avian species within the community pre- (2005–2010) and post-fire (2012–2021); we did not use relative abundance for this analysis. We tested for differences in the pre- and post-fire avian community groups using analysis of similarities (ANOSIM) [41,42] and investigated the species driving the distribution patterns pre- and post-fire in the NMDS using the function envfit in the package vegan v.2.6.8 [43]. Additionally, we used an Indicator Species Analysis using the function multipatt in the package indicspecies [44] to assess the differential occurrence of species in post-fire years. Before this multivariate analysis, we removed rare species from the avian community dataset; we defined rare species as those that occurred two times or fewer over the course of the 16-year sampling period. Due to the nature of migration and the chance for vagrants in our dataset we decided that two times over the course of the 16-year sampling period would remove most vagrants. We determined the appropriate number of dimensions by plotting final stress versus the number of dimensions and chose the number of axes beyond which reductions in stress were small [45].

To evaluate functional change in the avian community over our study period, we compiled trait data related to foraging and dispersal from AVONET [46]. We used bill length, bill width, bill depth, tarsus length, Kipp’s distance, hand–wing index, trophic niche, and lifestyle mode (aerial, perching, and generalist) as functional traits. We additionally included migration category (migrant, breeder, and resident) to calculate functional richness and identity for the full community. As with our NMDS, we removed rare species from the avian community dataset. For each study year and migration category, we generated a presence–absence matrix from which we constructed a matrix of functional Gower distances, which can handle both continuous and categorical data. We weighted all traits equally. Then, using principal coordinate analysis (PCoA), we constructed multidimensional functional spaces and assessed their quality using the quality.fspaces function from the R package mFD v.1.0.7 [47]. We selected the three PCoA dimensions with the mean absolute lowest deviation (MAD) and used those dimensions for calculation of functional richness, which measures the volume of the convex hulls of species trait space not weighted by abundance using the R package mFD v.1.0.7 [47]. Additionally, because culmen length correlated with the PCoA axis showing the smallest MAD, we plotted the average community culmen length over time (Appendix A).

To remove the effect of species richness on year-to-year community comparisons, we calculated standard effect sizes (SES) for functional richness [48]. We first constructed null community models (n = 1000) by randomizing traits among species but preserving trait covariance as in Ortega-Martínez et al. [49] and used these expected community values to calculate a standard effect size. We considered SES values statistically significant (α threshold < 0.05) if they fell outside the range of −1.96 to 1.96 standard deviations [50]. SES values that fell below −1.96 indicate under-dispersion and are thought to represent ecological filtering whereas values over 1.96 indicate over-dispersion and are thought to be connected to limiting similarity; values that fell within the range of −1.96 to 1.96 are assumed to represent stochastic processes [49].

We calculated the phylogenetic distance (PD) using the R package picante v.1.8.2 [51]. We downloaded 100 trees from birdtree.org using the Hackett backbone [52]. We pruned the phylogenetic trees to match our community data. We accounted for branch length and topological uncertainty by calculating PD across the entire posterior sample of 100 bird trees. For each topology and randomized presence–absence community matrix, we calculated the phylogenetic distance standard effect size (PD SES) by shuffling taxa tip labels (n = 1000) to generate null models. We visualized PD SES in ggplot2 v.3.5.1. We used the R statistical software v. 4.2.3 for all data analyses [53].

We tested for differences between pre- (2005–2010) and post- (2012–2021) fire years’ taxonomic, functional, and phylogenetic richness using a non-parametric Wilcoxon signed-rank test in R. To identify changes in species richness, SESs of functional richness, and SES PD associated with changes in PDSI and over time, we ran median-based linear models using the R package mblm v.0.12.1 [54]. The models are a robust form of regression and are less sensitive to data irregularities and outliers [54]. They implement a nonparametric Wilcoxon signed-rank test to evaluate significant relationships [54]. We used Spearman’s rank correlation to test for correlations between SES functional richness values.

## 3. Results

We banded 13,731 birds representing 77 species over the course of 16 years. Over the 16 sampling years, PDSI during the monsoon months significantly decreased, indicating increased drought over time during monsoon months (MAD = 0.41, V = 2666, *p* = 0.016), but showed no difference pre- and post-fire when tested as binary time bins (W = 32, *p* = 0.426; Appendix A). NDVI was significantly different pre- and post-fire when tested as binary time bins (W = 50, *p* = 0.001; Appendix A) and significantly dropped in the years post-wildfire (W = 40, *p* = 0.002; Appendix A). The results showed decreases in most categories of live vegetation post-fire except for young live trees (2.5–15 cm DBH; *p* = 0.021; Appendix A). We saw more young live trees in the 2.5–15 cm DBH category post-fire, and the majority were quaking aspen (Appendix A), although their distribution was variable on the landscape.

Although our dataset contained only 5 years pre-fire, our results showed some compelling patterns associated with wildfire and drought. Species richness showed a significant increase post-fire when tested as categorical time bins (W = 6, *p* = 0.023; Appendix A). Species richness significantly increased over the time period for all species (MAD = 0.42, V = 134, *p* = 0.0007; Figure 1A), even when broken into the subset categories of migrants (MAD = 0.20, V = 136, *p* = 0.0005), residents (MAD = 0.11, V = 119, *p* = 0.0009), and breeders (MAD = 0.23, V = 104, *p* = 0.002). However, species richness showed a significant negative relationship with PDSI but only for breeding birds (MAD = 0.74, V = 14, *p* = 0.030). Birds per 100 net hours increased immediately after the fire (Figure 1B) and significantly increased over time for breeders (MAD = 0.054, V = 119, *p* = 0.009) and migrants (MAD = 0.053, V = 111, *p* = 0.028). We did not detect a significant relationship (*p* > 0.05) between PDSI and changes in birds per 100 net hours over the length of the study period (Figure 1B).

NMDS results were generated from two convergent solutions and two dimensions, with stress = 0.129. We assessed differences in community composition and structure for pre- and post-fire years using the NMDS results (R = 0.67, *p* = 0.001). To investigate species driving the distribution patterns, we displayed vectors of species (*p* < 0.05) that showed strong influences on the community structure, all of which were associated with post-fire years (Figure 2a); these included Brewer’s sparrow, Cassin’s vireo, Cordilleran flycatcher, dark-eyed junco, downy woodpecker, Grace’s warbler, green-tailed towhee, lazuli bunting, mountain bluebird, olive-sided flycatcher, plumbeous vireo, red-breasted nuthatch, violet-green swallow, Virginia’s warbler, warbling vireo, white-breasted nuthatch, white-crowned sparrow, and western wood-pewee. The length of the vectors corresponds to the strength of the influence, with longer vectors having stronger influences and shorter vectors being less significant. To further assess specific species associated with post-fire conditions, we identified the differential occurrence of species in post-fire years using a species indicator analysis. The post-fire indicator species for our site are displayed with boxes around the four-letter species code in Figure 2a and included Virginia’s warbler, warbling vireo, plumbeous vireo, lazuli bunting, and Cassin’s vireo. Among bird species driving the difference between pre- and post-fire communities, the relative proportions of migrants and breeders showed a general increase in the years following the fire (Figure 2b) as did the relative proportions of insectivores and omnivores within the community (Figure 2c).

Functional diversity as measured by functional richness showed inconsistent trends over time and with climate. Three traits were significantly correlated with the PCoA axes showing the lowest MAD: posterior and anterior bill length and feeding guild (Appendix A). Only a single SES functional richness value significantly differed from random expectations and all values showed no temporal trends for the overall community or any subset (Appendix A). Only resident species in 2006 showed significantly lower than expected values of SES functional richness. All other annual standard effect size estimates fell within null expectations. The SES functional richness values showed significant associations with time. The overall community SES functional richness declined over time (MAD = 0.207, V = 2167, *p* < 0.001; Appendix A), running counter to the trend in species richness and SES functional richness of community components. SES functional richness significantly increased over time for the migratory, breeding, and resident subsets of the community (migrant: MAD = 0.125, V = 4924, *p* < 0.001; breeder: MAD = 0.156, V = 4789, *p* = 0.002; resident: MAD = 0.185, V = 4436, *p* = 0.035; Appendix A). SES functional richness of the overall community and breeding component subset showed no differences between pre- and post-fire years (overall community: W = 29, *p* = 0.665; breeders: W = 39, *p* = 0.093). SES of functional richness of the migratory and resident components showed significant increases post-fire (migratory: W = 48, *p* = 0.006; resident: W = 43, *p* = 0.031). The SES functional richness of the overall community was not associated with mean monsoon PDSI (MAD = 0.845, V = 3183, *p* = 0.4305; Appendix A). However, the functional richness of resident and migratory species showed significant negative relationships with decreasing drought (migrant: MAD = 0.501, V = 2658, *p* = 0.016, resident: MAD = 0.742, V = 2594, *p* = 0.010; Appendix A). The average culmen length of the overall bird community decreased in the years following the fire, shifting toward a community with shorter bills (Figure 3a), but the difference between pre- and post-fire years was only marginally significant (W = 4, *p* = 0.052). Only 3 years showed significant SES PD values. The SES PD in 2009 was higher than null expectations, whereas 2008 and 2019 were significantly lower than null expectations (Figure 3b). The trend in SES PD fluctuated over time but showed a significant decreasing trend (MAD = 0.3546, V = 2169, *p* < 0.001). SES PD also showed a weaker significant negative association with mean monsoon PDSI (MAD = 1.07, V = 2093, *p* = 0.0276). The SES PD showed a strong positive correlation with average culmen length in the community (MAD = 2.43, V = 5540, *p* < 0.001; Figure 3).

## 4. Discussion

The duration of extreme droughts and instances and intensity of wildfire are expected to increase. Birds will need to adapt to keep up with changing resource bases within their breeding habitats and along their migratory pathways [55]. Our study illustrates the associations along multiple axes of biological variation of both wildfire and drought on an avian community during fall migration.

Wildfires can act to reset communities to varying degrees, depending on their severity and/or extent. Initially, wildfires can deplete long-standing resources for higher-order consumers. But over time, competition and an influx of carbon and nitrogen can drive rapid succession in the community of primary producers [56,57,58]. Unlike the pulse effects wildfire can have, drought operates over longer timescales. The dichotomy between short- and long-duration ecosystem impacts has been framed as pulse–press dynamics [59,60].

Pulse-like, nonlinear changes in avian abundance and species diversity indices following wildfire have been documented for breeding communities [61]. Our results show similar short-duration shifts in a migratory stopover community post-fire. We saw the overall species richness (for all categories) increased over time (Figure 1A), and an increase in birds caught per net hour for migrants and breeders at the site following the 2011 wildfire (Figure 1B), presumably due to the changes in habitat created after the fire. This reorganization is reflected in species-level shifts in relative abundance and the overall community structure pre- and post-fire (Figure 1 and Figure 2) and could be driven in part by mismatches among the ecological requirements of species, post-fire resource availability, and habitat conditions, forcing emigration or discouraging use of the area as a stopover site. The post-fire vegetation shift to dense but patchy areas of quaking aspen cover was reflected by the drop in September NDVI values at the banding site after 2011. Unlike mature conifers, young deciduous trees like aspens have lower canopy structure, chlorophyll density, and overall leaf area, each of which can lead to lower NDVI values [62,63] (Appendix A). These same changes in the habitat structure and net primary productivity of the post-fire successional plant community (Appendix A) could have led to functional changes in the avian community by attracting species that would normally have bypassed the habitat [64]. For example, the increased cover provided by patchy areas of young, dense, early successional quaking aspens (Appendix A) could attract migratory avian insectivores by providing increased high-density cover from predators and the elements and by promoting insect abundance [65,66]. We saw this pattern emerge in our dataset where insectivorous migratory species were documented more often in post-fire years (Figure 2b,c). The species identified in our indicator analysis for post-fire conditions also suggested this trend because all were migratory insectivores (Figure 2a).

Recent studies have highlighted how the foraging behavior of birds influences their post-fire occurrence. Birds that forage on open ground were more abundant in the years soon after fire [67], whereas birds that depend on well-developed, midstory structure were more abundant in later successional vegetation [68]. At our study site, the chipping sparrow (*Spizella passerina*) —a ground forager—showed one of the highest relative abundances immediately after the wildfire during the same year, and Wilson’s warbler showed one of the highest relative abundances in later years post-fire (Appendix A). Irruptive patterns have been recorded in Spizella sparrows [69], but we are skeptical that irruptive migration drove the dominance of chipping sparrows in 2011 because their numbers were still relatively low compared with other years pre- and post-fire.

Multiple community properties showed associations with both fire and drought. Among species driving pre- and post-fire community separation, the relative abundance of migratory insectivores increased in the years immediately following the fire (Figure 2). This increase coincided with a shift toward a more related, less phylogenetically dispersed community (Figure 3b), which coincided with a shift in apparent community function toward shorter-billed birds (Figure 3a). The initial post-fire shift could be the result of high temporal turnover while species less likely to forage in mixed-conifer forests investigate newly structured and homogenized habitat during migration. Functional richness showed no clear association with wildfire. While functional richness increased significantly over time in the migratory, breeding, and resident components of the community, only the migratory and resident components had significantly higher diversity during post-fire years; this could reflect post-fire vegetation changes attracting migratory insectivores. However, we urge caution interpreting the functional richness metrics since nearly all values fell within significance thresholds, indicating that most years were in line with expectations given stochastic community assembly. Furthermore, overall functional richness showed an opposing pattern relative to the community components, illustrating how Simpson’s paradox and our small samples sizes may hinder the interpretation of component trends of functional richness [70]. The breeding community subset may also be absorbing statistical power from our migrant group, because all birds from this category are long-distance migrants that may be utilizing the local habitat for longer periods of time. More compelling, however, was the overall community relationship between mean culmen length and phylogenetic diversity suggesting a decrease in functional diversity—with respect to this foraging trait—of the community post-wildfire (Appendix A). It is possible that other measures of functional diversity might capture signatures of wildfire, but we restricted our analyses to functional richness and average culmen length because neither are abundance weighted, and mist-netting data are thought to bias abundance estimates within bird communities [38,71].

The more pervasive and subtle effects of drought have downstream effects on all ecosystem services and could be responsible for slow habitat degradation along migratory routes. Many migratory birds track environmental conditions on their breeding and non-breeding grounds [72,73], but much less is known about how migratory birds select stopover habitat [74,75,76,77]. Drought is known to negatively impact birds on their breeding grounds [7,78]. It stands to reason that drought-degraded habitat could cost migratory birds time and energy, as well as influence migratory bird abundance [8]. Running counter to our expectations, we found that in years when the monsoon produced wetter conditions, functional richness of resident species and migrant species, overall community phylogenetic diversity, and breeding bird species richness decreased. The precipitation-driven habitat shifts in relation to a wetter monsoon could favor certain groups of birds, leading to declines in species and functional richness, as well as phylogenetic distance at a local scale. For example, resource pulses following the monsoon season could alter local and long-distance migration phenology for all species by directly influencing the timing and availability of food sources. Alternatively, because wetter monsoon years were mostly pre-fire (PDSI monsoon season median = 0.050), the relationship may be driven by post-fire condition effects on increased species diversity which also happened to be the drier years (PDSI monsoon season median = −0.715) in our relatively small dataset (Appendix A). Ultimately parsing the effects of long-term drought and wildfire was not possible with our data from a single site and small sample sizes.

Overall, we found some evidence that the wildfire-changed habitat at our study site attracted and supported new species combinations during migration and shifted community composition. However, the taxonomic, functional, and phylogenetic shifts that follow fire could be variable and short-lived. Birds’ capability to select habitats during migration will hinge on the availability of adequate stopover sites, highlighting the growing importance of preserving productive microhabitats in a drought-stricken region. Moving forward, researchers should expand on our localized analyses and leverage open-access banding data and occurrence records to model the associations between taxonomic, functional, and phylogenetic diversity and pulse–press disturbances such as wildfire and drought at a continental scale. Larger studies at regional or continental scales could leverage the wealth of open-access data to tease apart the effects of wildfire and drought with more complex and powerful modeling approaches.

## 5. Conclusions

While instances of extreme drought and wildfire occurrences and severity increase, migratory bird communities could shift to alternate areas and migratory routes that have the resources available to support their needs. Our results suggest that migratory birds can rapidly respond to changing resources along their migratory pathways to find suitable environmental conditions as they move across the landscape. However, facilitating conservation through site identification and protection, such as using wildland fire forest management and the conservation and creation of areas that provide stopover refuge, is increasingly essential in the protection of migratory birds during this crucial time in their annual life cycle—when refuge could mean the difference between a successful migration and death.

## Figures and Tables

**Figure 1 biology-14-00597-f001:**
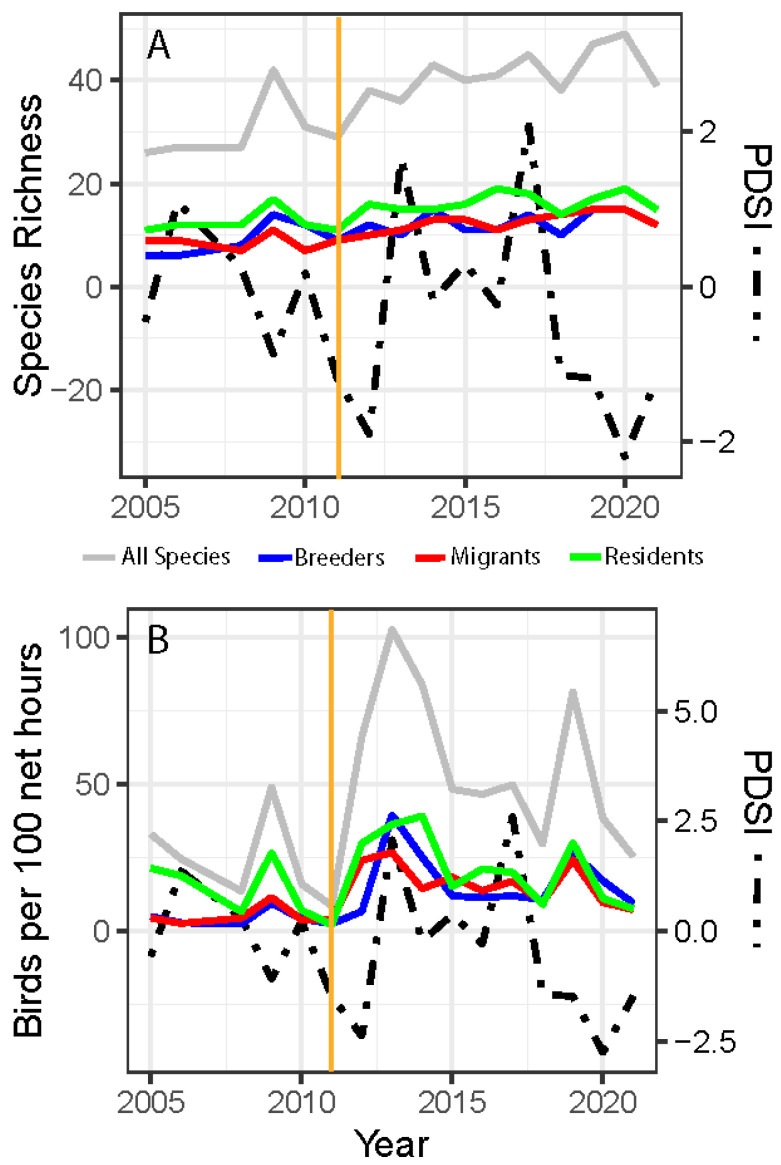
Species richness (**A**) and birds per 100 net hours (**B**) displayed on the left *y*-axis and average monsoon (June–September) PDSI values on the right *y*-axis from 2005, 2006, and 2008 to 2021; all species are shown with a gray line, residents are shown in green, migrants are shown in red, and breeders are shown in blue; orange vertical lines indicate the fire event.

**Figure 2 biology-14-00597-f002:**
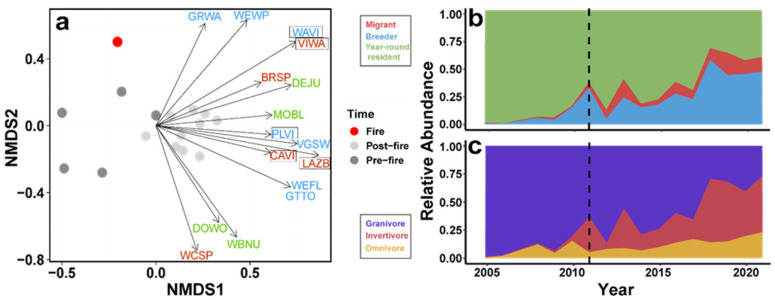
Avian community composition over time. (**a**) Nonmetric multidimensional scaling (NMDS) of the avian community pre- (2005, 2006, and 2008–2010) and post-fire (2012–2021; R = 0.69, *p* = 0.003). Dark gray points represent pre-fire years, the red point indicates the year of fire, and light gray points indicate post-fire years. Vector arrows show significant species colored by migratory category and the direction of influence driving the distribution patterns pre- and post-fire in the NMDS (*p* < 0.05); species with boxes around them are indicator species for post-fire conditions (*p* < 0.05). BRSP = Brewer’s sparrow, CAVI = Cassin’s vireo, WEFL = western flycatcher, DEJU = dark-eyed junco, DOWO = downy woodpecker, GRWA = Grace’s warbler, GTTO = green-tailed towhee, LAZB = lazuli bunting, MOBL = mountain bluebird, PLVI = plumbeous vireo, VGSW = violet-green swallow, VIWA = Virginia’s warbler, WAVI = warbling vireo, WBNU = white-breasted nuthatch, WCSP = white-crowned sparrow, WEWP = western wood-pewee. (**b**) Area plot showing relative abundance of migratory groups of significant NMDS species over time. (**c**) Area plot showing relative abundance of trophic groups of significant NMDS species over time. Relative widths of color bands indicate shifts in proportions of migratory and foraging modes.

**Figure 3 biology-14-00597-f003:**
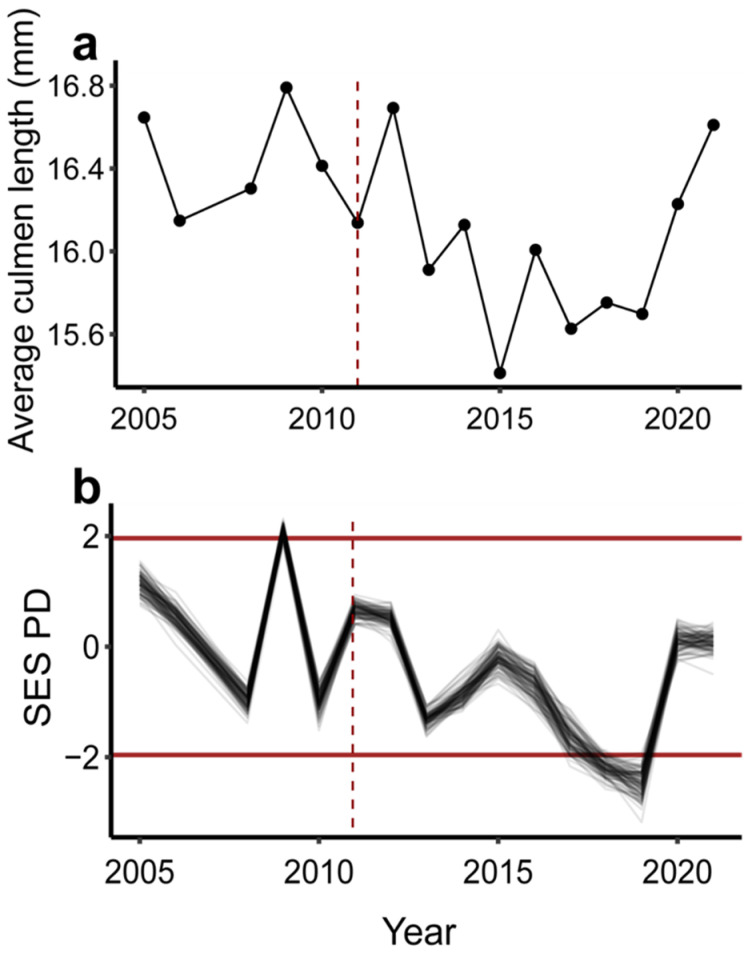
Functional change, as measured by average community culmen length, and phylogenetic distance over time. (**a**) Average community culmen length over time. Points represent the mean culmen length of the overall community each year. The black line connects points to emphasize chronology. The red dashed vertical line indicates year of fire. (**b**) Standard effect sizes for phylogenetic distance (SES PD) over time. Black lines connect yearly SES PD values. Each line represents yearly SES PD values for 1 of 100 different phylogenetic tree topologies. Red horizontal dark lines indicate significance thresholds for SES PD values. Red dashed vertical line indicates year of fire.

## Data Availability

Publicly available climate datasets were analyzed in this study. These data can be found here: https://www.ncdc.noaa.gov/cag/ (accessed on 11 April 2025). Publicly available avian trait data related to foraging and dispersal from the AVONET database were analyzed in this study and can be found here: https://opentraits.org/datasets/avonet.html (accessed on 10 April 2025). The original contributions presented in this study are included in the article/Appendix A. Further inquiries can be directed to the corresponding author.

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
