# Peer review of "Fire and Drought Affect Multiple Aspects of Diversity in a Migratory Bird Stopover Community"

_biology, 2025, doi:10.3390/biology14060597_

Round 1

Reviewer 1 Report

Comments and Suggestions for Authors

Congratulations for your study, it is very well done and written. Only one sugestion:

"The fall migration monitoring site, Upper Alamo in Bandelier National Monument, 121 is located at the Alamo Boundary trailhead (35.834 N, 106.608 W) in the northwestern area 122 of Bandelier" - State and country?

And did you find any species that benefit from the fire? It is something that we can see in other areas mainly with raptors?

Reviewer 2 Report

Comments and Suggestions for Authors

The research uses a bird-banding dataset to evaluate avian community shifts in a drought area following wildfire. The overall results suggest that while birds are immediately affected by fire-driven resource changes, they can recover over time. However, the long-term effects of drought on the composition of avian communities remain uncertain. I found the text is well-structured and the core argument, linking drought, fire, and avian community responses, is compelling and logically supported. I only have a few minor points for the authors to revise.

Lines 75-76: "pushed to the edge of their physiological limits [20] and can experience the largest amount of annual mortality". Please rephrase it to: "pushed to their physiological limits [20] and experience the highest annual mortality rates..."

Line 101: delete the “or” before local extinctions.

Line 180: Can you explain why “you decided that the annual monsoon values would be more indicative of the conditions during which birds were captured and data were collected”?

Lines 245-249: The information should be placed in the Method section.

Line 259: Here, figure 1a was cited to show “species richness showing a significant negative relationship with PDSI but only for breeding birds”. However, figure 1a is hard to interpret and from the black dotted line, one can not tell only breeding birds’ richness had a significant negative relationship with PDSI.

Lines 285-287: “the relative proportions of migrants and breeders showed a general increase in the years following the fire (Figure 2b) as did the relative proportions of insectivores and omnivores within the community (Figure 2c).” I cannot tell pattern from the figures.

Comments on the Quality of English Language

The English is overall good, but can be improved.

Reviewer 3 Report

Comments and Suggestions for Authors

Stenek et al.  

Long-term population studies are scarce because it’s hard to find money and personnel to keep them going year after year.  Yet, ecological processes are often punctuated by severe episodic events. In dryland ecosystems, fires and droughts are such examples.  They can have dramatic, long-lasting impacts, nonetheless,  ones hard to quantify without years of data.  On top of that most studies are of breeding populations since they are often easier to count.  All this against a background of well-established trends in climate disruption.  

In short, this is an important, exceptional paper, to which I say “very well done.”  I have nothing substantial to add by way of comments. The methods are straightforward, just hard work year after year. 

In case some reviewer gets snippy about drawing conclusions about one place and basically one event, that’s just the nature of nature in the Southwest.  Some 350 km to the south, in an age when dinosaurs were a threat to fieldwork, I did a year-round bird study for a few years, during IBP.  Just how much things can change from year to hard to imagine for those not familiar with these ecosystems.  Dryland ecosystems are exceptionally dynamic and this adds a well-documented example.

Reviewer 4 Report

Comments and Suggestions for Authors

This study uniquely examines the combined effects of fire (a short-term "pulse" disturbance) and drought (a long-term "press" disturbance) on avian communities during migration. While prior work focuses on breeding communities, this manuscript addresses a gap by analyzing stopover dynamics, integrating taxonomic + functional + phylogenetic diversity metrics, and disentangling responses across migratory strategies.

Major Concerns

1.​ Causality vs. Correlation: Limited vegetation data (pre/post-fire snapshots) and lack of continuous habitat metrics weaken claims about fire-driven mechanisms.

2. Temporal Confounding: Post-fire years overlap with ongoing drought; disentangling their effects is incomplete. The discussion overattributes shifts to fire without robustly addressing drought’s compounding role.

 Specific Comments(I couldn't find the line number)

  1. Removing rare species (≤2 occurrences) risks omitting fire-sensitive taxa. Sensitivity analyses (e.g., including/excluding rare species) would strengthen conclusions.
  2. SES functional richness results (mostly non-significant) contradict claims of drought-driven functional shifts. The emphasis on significant trends in subsets (migrants/residents) risks overinterpreting noise.
  3. Reassess drought-fire interactions using multivariate models (e.g., PDSI × time since fire)
  4. Missing key studies on fire/drought interactions in avian communities (e.g., Jones et al. 2023, Ecological Applications)

Decision Recommendation: Minor 

The study addresses an important gap with novel multi-dimensional analyses and long-term data. However, methodological limitations (vegetation data, sample size, confounding factors) and overinterpretation of non-significant trends require revision. This work could provide valuable insights for conservationists managing migratory stopover habitats in fire/drought-prone regions after addressing problems above.

Reviewer 5 Report

Comments and Suggestions for Authors

This study examined how wildfire and drought shape avian relative abundance, taxonomic, functional, and phylogenetic diversity using fall bird-banding data. It is an interesting article with a large database and I consider it suitable for publication in biology, the text would be improved with these two suggestions:

Line 49 change "American South west" to "Southwest of the USA".

Line 382 put the scicentific name of the chipping sparrow.

Author Response

This study examined how wildfire and drought shape avian relative abundance, taxonomic,  functional, and phylogenetic diversity using fall bird-banding data. It is an interesting article with a  large database and I consider it suitable for publication in biology, the text would be improved with  these two suggestions: 

Line 49 change "American South west" to "Southwest of the USA". 
Response: Updated to the suggested change.

Line 382 put the scientific name of the chipping sparrow.
Response: Added the scientific name of the chipping sparrow.